# Interactive, computer-based, and situated design for innovative formative assessment approaches

Federica Morleo[1,2*], Pasquale Anselmi[1], Alessandra Vitanza[2]

**1** University of Padua, Department of Philosophy, Sociology, Education and Applied Psychology (FISPPA), Padova, Italy, **2** Institute of Cognitive Sciences and Technologies, National Research Council (CNR-ISTC), Catania, Italy

* federica.morleo@phd.unipd.it

## Abstract

Social and teamwork skills are essential for today's teachers, yet their assessment in authentic contexts is challenging. This study presents the design, development and validation of an innovative computer-based test, designed to assess collaborative and teamwork skills in preprimary and primary school teachers and referred to as the 'CoTeSt' (Collaborative and Teamwork Skill test). Based on Situated Action Theory, the test immerses participants in realistic team interactions using conversational agents within a narrative framework. Teachers are asked to solve problems, collaborate with virtual colleagues, and provide feedback. The test consists of 20 dichotomously scored items in both multiple-choice and short answer, and it was administered to 139 Italian teachers. Test validation involved qualitative and quantitative methods, confirming that the items actually assess the skills they were assumed to evaluate. Post-test interviews and group discussions highlighted the tool's user-friendly design and its potential to foster self-reflection, professional dialogue, and continuous skill development. The CoTeSt represents a meaningful step toward empowering teachers with critical social skills and fostering a culture of collaboration and growth in education.

## Introduction

Today, the role of teachers is increasingly recognized as essential to society worldwide, highlighting the importance of their high-quality professionalism [1,2]. Despite this recognition, many challenges remain because teachers' status is not universally acknowledged as part of higher education (HE). Indeed, as pointed out by the OECD Review of School Resources [3], the qualification requirements for primary education teachers, as well as school leaders, vary considerably from country to country. Nevertheless, it is broadly acknowledged that teacher education and training should be based on autonomy and responsibility, connections with evidence-based research, and strong ties to society [2]. Therefore, their training is crucial for developing a range of attitudes, values, and behaviors necessary to provide appropriate competencies for all learners [4]. The importance of teacher and teaching quality has been highlighted by the Teaching and Learning International Survey (TALIS) [2] and, more

**Data availability statement:** The CoTeSt (both in English and Italian versions) and the data analyzed in this work are publicly available at the following link: https://osf.io/57e6w/?view_only=d5618800397e41c4bda4cb3cb590e394.

**Funding:** The author(s) received no specific funding for this work.

**Competing interests:** The authors have declared that no competing interests exist.

recently, by the OECD project [5]. They identify five key elements of teacher professionalism, both individual and collective, that integrate diverse perspectives. Specifically, these elements include the development of a knowledge and skills base, access to quality training and lifelong learning, opportunities to foster collaboration, the power to be responsible and autonomous in policy and practice, and the prestige of the profession. Indeed, the framework outlines various ways to enhance teacher quality, with one based on promoting collaboration. The idea of teacher collaboration and teamwork with colleagues, school headteachers, students, families and external educational stakeholders is strongly related to school improvement and effectiveness; indeed, they are considered essential pillars of teacher professionalism [6]. Collaboration requires skills to find solutions to complex challenges, engage with networks, and manage work across organizational boundaries [7]. In today's educational context, there is a trend toward cooperative learning to meet society's workforce needs. Skills such as perspective-taking, conflict resolution, negotiation, and the ability to promote inclusive environments are just a few examples within the framework of collaboration and cooperation. These skills are especially relevant for teachers, as they can influence the development of students' collaborative skills [8], another important goal for the future of education. Despite different understandings of skills and competences around the world, this paper adopts the European Qualifications Framework definition of competence as the ability to 'know how to act' in a specific context [9]. In particular, taking into account the contribution of Le Boterf, a competence can be traced from the knowledge of 'how to do' to 'how to act' [10]. According to this model, three levels are essential for analyzing competences: Knowledge Resources (i.e., contents of knowledge), Cognitive and Operative Processes (i.e., actions required to solve problems), and Dispositions to Act (i.e., affecting behavior management in the context) [11]. Since the emergence of the concept of competence, different approaches have been developed and globally it is possible to find the same meaning under the name of 'skill' due to the spread of Life Skills [12]. The first step toward effectively improving specific skills is assessment, which provides a clear understanding of strengths and weaknesses and offers precise feedback to guide interventions. Adapting learning processes based on assessment outcomes involves moving towards a formative assessment, which aims to support learners [13]. This is a powerful approach for evaluating learners' knowledge and skills to foster further development [14]. Indeed, the key element of formative assessment lies in understanding the information received to improve specific abilities: the language and the content need to be clear to learners. To be effective, this process should be supported by a range of strategies that promote improvement [15]. In psychology, an innovative approach to assessment is Knowledge Space Theory (KST) [16]. Unlike traditional approaches (e.g., classical test theory, item response theory), which offer a numerical representation of an individual's knowledge level in a specific subject area, KST provides a discrete, non-numerical yet precise description of the specific knowledge an individual possesses in that area, in line with the aims of formative assessment. Competence-based Knowledge Space Theory (CbKST) [17,18] is an extension of KST designed to meet the needs of educational systems by identifying the specific skills that an individual already masters (i.e., the individual's competence state) and skills they are ready to learn (i.e., the outer fringe of the individual's competence state). In CbKST, the outer fringe of the competence state formally implements Vygotsky's [19] concept of the zone of proximal development. In contexts where individuals interact with others, they can progressively conquer their zone of proximal development [18]. CbKST has been successfully applied to developing tests for various assessments, including basic statistics [20–22], arithmetic problem-solving [23], and fluid intelligence [24,25]. Within CbKST, tests can be constructed to be short yet maximally informative about the skills individuals master [26,27].

In this paper, the design of an experience aimed at assessing the collaboration and teamwork skills of preprimary and primary school teachers is proposed in response to significant challenges [28,29]. Section Challenges and Our Proposals examines three key challenges from a theoretical perspective and addresses each of them with the corresponding practical solution adopted in this work. Finally, section Results presents the results obtained on the CoTeSt (**Co**llaborative and **Te**amwork **S**kill **t**est), which is the test for assessing teachers' collaborative and teamwork skills developed in this work.

## Challenges and our proposals

### Challenge 1: Defining collaborative and teamwork skills

Collaborative and teamwork skills empower individuals to better address challenges and needs in their lives. These multidimensional skills encompass knowledge, behaviors, and attitudes that promote resilience and adaptability [30]. In order to define teamwork skills, the starting point is to provide a definition of the term 'team'. Following the Kozlowski and Bell [31] definition: "*A team is composed of two or more individuals who (a) exist to perform organizationally relevant tasks, (b) share one or more common goals, (c) interact socially, (d) exhibit task interdependencies, (e) maintain and manage boundaries, and (f) are embedded in an organizational context that sets boundaries, constrains the team, and influences exchanges with other units in the broader entity*". Team members' characteristics play an important role, as they influence both teamwork processes and performance [32]. Indeed, teamwork processes concern how individuals combine their personal skills to solve a specific task [33]. In this sense, each member contributes to the team through their personal skills by impacting the overall success of the task [34–36]. Within this framework, Stevens and Campion [35] identified five essential skills for effective teamwork, with the aim of capturing individuals with high levels of these abilities. They are grouped into two domains: interpersonal skills (conflict resolution, collaborative problem-solving, and communication) and self-management skills (goal setting, performance management, planning, and task coordination) [37]. Cannon-Bowser et al. [36], a year later, proposed eight individual skills related to teamwork effectiveness: adaptability, shared situational awareness, performance monitoring and feedback, leadership and team management, interpersonal relations, coordination, communication, and decision-making. Salas et al. [38] further categorized teamwork and collaboration into three types of processes: transition (e.g., goal setting, strategy formulation), action (e.g., coordination, monitoring) and interpersonal relations (e.g., motivation, conflict management). Moreover, Petkova et al. [32] described teamwork skills as a result of effective interaction, empathy, communication, time management and decision-making. Similarly, Rodrìguez-Sabiote et al. [39] characterized teamwork skills by different components, reflected in the perceptions toward the team, the interaction among members and the execution of the task. The Partnership for 21st Century Skills [40] also provided an overarching view of collaborative skills, including group decision-making, interpersonal and communication skills, and group support, with the PISA (Programme for International Student Assessment) framework adding reflection and monitoring processes [41]. In this context, skills are used as a synonym for competence. In line with this exploration, the teamwork skills identified during the development of the CoTeSt are discussed below.

**Assessing collaborative and teamwork skills in teachers.** Although it is difficult to describe the multifaceted nature of teamwork skills, two domains are commonly identified across various studies on collaborative and teamwork skills, as discussed in the previous section. The first domain is related to interpersonal relationships, and the second to problem-solving processes. The interpersonal domain includes skills like empathy, communication,

and getting along well with others [42], while the problem-solving domain encompasses skills such as decision-making, process control and adjustment, responsibility, and perseverance [43]. Therefore, in developing our assessment tool, we considered both the *relationship* and *work management* domains. The difficulty in defining collaborative and teamwork skills is reflected in the limited international research on learning and assessing life skills [44]. Indeed, current studies are typically based on observational research, focusing on how individuals, especially students, demonstrate the development and acquisition of these skills [45]. The CoTeSt focuses on two subdomains in the *relationship* domain: 'effective communication' and 'conflict resolution'. In the *work management* domain, attention is given to 'action planning' and 'performance monitoring and feedback'. Furthermore, the choice of the skills to be assessed is driven by the need to focus on those that can be observed in the individual's behaviors alone, although they involve interaction with others. Exploring collaboration and teamwork skills for teachers is challenging because teaching is a multifaceted profession. However, TALIS provides a broad perspective on teacher teamwork activities and outlines what teachers need to promote collaboration in their practice. Indeed, teachers have many opportunities to interact and collaborate with others, both formally and informally, at different levels of interdependence and coordination [8]. The six collaborative and teamwork skills selected for the CoTeSt are provided in Table 1, alongside a description of their application in teacher professionalism.

## Challenge 2: Teacher assessment

An international study conducted by OECD [46] and involving twenty-five educational systems around the world found that the 'teacher quality' is the most important variable within the school context in promoting school effectiveness. However, there is a lack of consensus in the literature on its definition and practical significance, which leads to confusion about *what* and *how* to assess in teachers [47]. Research studies about teacher quality are typically focused on students' achievements [48] as an outcome of teacher effectiveness. Other studies evaluate teacher quality using tests, observational protocols in the classroom, and questionnaires, attempting to measure characteristics like salary, career progression opportunities, and discipline knowledge. However, these studies neglect the elements that are harder to assess within an evaluative environment, such as the ability to collaborate with colleagues, create effective relationships, and guide self improvement. These skills are considered vital for students' learning as well. Indeed, students learn not only from the teachers, but also through interaction with peers [49], so fostering a collaborative learning environment is crucial. Beyond 'academic' goals, primary education plays an important role in helping students build the foundation for learning that includes a wide range of social and emotional skills, along with collaboration and cooperation with others [50]. Therefore, the type of teaching required in primary schools differs from that in other education levels and requires teachers to have specific competences able to stimulate students' transversal skills. As outlined by OECD [7], these competences should be assessed both during teacher training and throughout their careers to enable relevant and engaging professional development within schools. To date, the TALIS study highlights the difficulty of measuring teachers' professional collaboration, despite encouraging its promotion from the earliest education levels [2]. Reflecting on these observations, a proposal for creating an environment where teachers' skills can be evaluated is presented in the following section.

**Online assessment.**   Collaborative and teamwork skills are part of an interactive, complex process that is challenging for an accurate and consistent measurement in teachers, as outlined in the previous section. The complexity increases when the aim is to create a realistic

**Table 1. Collaborative and teamwork skills in teacher professionalism selected for the CoTeSt.**

| Domains | Subdomains | Skills | Applications |
|---|---|---|---|
| Relationship | Effective Communication | Using proactive communication strategies | Proposing weekly tasks updates Using verbal and non verbal communication with students |
| | | Using active listening techniques | Paraphrasing a colleague's idea Writing key concepts during a brainstorming session |
| | Conflict Resolution | Recognizing type and relevance of conflict situations | Identifying role confusion in group tasks Managing peer conflicts among teachers during group activities |
| | | Identifying conflict resolution strategies | Suggesting and encouraging collaborative problem-solving among students Managing and balancing the workload among colleagues |
| Work Management | Action Planning | Identifying and planning action strategies | Developing group task timelines Identifying a shared objective that a group of students are required to achieve Assigning roles based on the skills possessed by individuals Focusing on completing a project or task |
| | Performance Monitoring and Feedback | Monitoring performance and providing feedback | Identifying unclear instructions given to students Highlighting students' strengths and weakness Suggesting targeted pedagogical strategies to colleagues Recognizing obstacles or errors in a colleague's classroom performance |

environment in which these skills are applied. In order to control the assessment contexts and to collect and analyze user performance, a computer-based assessment was chosen. The assessment process is interactive, allowing teachers' collaborative skills to be explored by isolating the distinctive aspects that are individually evaluated. The assessment assumes that almost all preprimary and primary school teachers in 2024 would be familiar with computers, especially because information and communications technologies (ICT) become more widespread in our society and in schools [51]. Teachers are now required to use digital school records to track student progress, record lessons, and share homework or exercises with families and the wider school community.

**Scenario-based assessment.** The entire test is drawn on a scenario-based assessment, theoretically grounded in Situated Action Theory [52,53]. This implies the presence of specific circumstances, where planned actions in human-computer interactions occur. This context is provided through a narrative framework that immerses teachers in a story where they have to engage in team discussions, interact with students, or collaborate with colleagues on joint activities and provide feedback. In other words, the context mirrors real job situations, allowing teachers to demonstrate their collaborative and teamwork skills in interactions with computer agents. The job situations explored in TALIS [8] reflect different opportunities for teachers to use their collaborative skills, such as interacting with colleagues and school leaders and making decisions to solve problems in their practice. In line with the findings in TALIS,

we should mention: (a) teaching in a team, (b) giving feedback, (c) engaging in joint activities across classes, (d) participating in collaborative professional learning, (e) sharing materials, (f) discussing students' learning progresses, (g) working with other teachers to ensure common evaluation standards, (h) attending team meetings, and (i) participating in school collegiality. The scenario designed takes into account the types of challenges teachers may face in the context of their work, including interactions with colleagues, head teacher, and students. It involves multiple contexts in which teachers employ their collaborative and teamwork skills in everyday professional life, categorized into two dimensions: (a) engaging in team discussions about classes with other teachers, and (b) collaborating with colleagues on joint activities and providing feedback. Each dimension covers the range of skills involved in collaborative and teamwork processes, which differ in the contexts in which they are situated in the field. For instance, in a consensus-building task, a team of teachers may reach a common decision on how to manage the participation in a national call for proposal, when each has their own ideas, knowledge, and didactic methods. The choice of a scenario-based assessment also arises from the need to create an engaging and motivating testing environment that immerses teachers in the narrative framework. This approach allows teachers to respond based on their genuine thoughts rather than socially desirable answers.

**Conversational agents.**  Collaborative and teamwork skills are assessed by examining teachers' behaviors during interactions with one or more agents. The test provides an accurate measure of a teacher's skills without relying on the performance of others. In fact, by using computer simulations of human agents, the test can control the feedback provided by computer agents regardless of respondents' choices. Since the focus is on evaluating the skills of individual teachers through their responses, the contribution of the rest of the group is planned in advance and thus controlled. In real life, teachers must be prepared to work with various and heterogeneous individuals, as some may collaborate more effectively with specific people than with others. Thus, the only way to obtain a comprehensive and valid estimate of teachers' collaborative and teamwork skills was to create team members with different skills' proficiency to interact with. Teachers interact with these agents through a chat, which ensures a high level of control and great engagement. The responses of conversational agents are pre-defined, contextually aligned and coherent with any of the alternative choices selected by teachers. This ensures that all participants are presented with the same set of items. With regard to this aspect, the OECD [41] findings indicate that examinees' performance, as measured by task scores, is comparable between computer-based tasks and human interactions. This suggests that well-structured, pre-defined interactions mediated by agents (e.g., through on-screen messages) can effectively simulate the conditions of human interaction within the assessment, ensuring coherence and contextual relevance.

## Challenge 3: Formative assessment

Recently, increasing worldwide attention has been directed toward 'formative assessment', also known as 'assessment for learning', which supports individual development through feedback [54]. In this sense, assessment should be understood as a form of communication, capable of encouraging individuals' improvement. However, implementing formative assessment is challenging, primarily because its meaning varies across countries, school systems, and stakeholders. In contrast to 'summative assessment' or 'assessment of learning', formative assessment offers a continuous and precise description of individuals, enabling them to act on this information. The term 'formative' implies more than simply the frequency and

timing of the assessment alongside the teaching practice. It needs to support learning, informing both students and the teachers on how to develop further learning [55]. Rather than providing a singular definition that encompasses the various ways in which formative assessment is conceptualized, it may be more fruitful to focus the attention on the actions that must be present within a formative assessment. These actions are conceptualized as the empirical evidence that can be observed when practicing formative assessment. In this view, The Assessment Reform Group [56] identifies three core actions related to formative assessment: 'seeking', 'interpreting', and 'acting upon interpretation'. Bennett [13] also adds the 'planning and designing the assessment act'. Each act involves multiple interpretations, such as understanding the curriculum, classroom social dynamics, or teacher autonomy. For instance, within the 'seeking' assessment act, one should determine what needs to be assessed, while in the 'planning and designing the assessment act', one should select the best assessment tools based on the aims, the participants, and the relevant personal skills, such as technological skills, if needed. Research indicates formative assessment is an effective way for improving learning outcomes, as outlined by Black and Wiliam [57], Bloom [58], Hattie [59,60], and the Centre for Evaluation and Monitoring [61]. However, understanding how formative assessment can be practiced effectively is crucial. According to the previously mentioned acts, four core aspects support the practice of formative assessment. The first one is the 'awareness of the moment' [62], which is relevant to both the evaluator and the person being evaluated. Awareness is related to both cognitive aspects of the evaluation, such as knowledge or skills, and experiential factors like motivation, self-esteem, and communication. The second aspect is the 'type of questions' [63], which should offer multiple sets of questions and answers in a specific content domain. The third core aspect is the 'zone of proximal development' [19], which, in the framework of formative assessment, refers to the self-realization that individuals, after trying the assessment experience, may well assess themself on their own because they have understood the goal of the assessment [64]. The fourth concept emphasizes the role of the 'community of practice' [65] within each assessment context, which is characterized by its own norms and culture. These factors influence how assessment is perceived, as a support or a judgment, and how the environment is perceived as safe and secure or threatening. Our test was designed using a specific formal approach that emphasizes the core aspects of formative assessment, as described in the next section.

**Competence-based knowledge space theory.** Competence-based Knowledge Space Theory (CbKST) [17,18] is a mathematical theory for the formative assessment of skills that aims to precisely identify the skills mastered by an individual (i.e., the competence state of the individual) based on the observed responses to test items. Examples of application can be found in Anselmi et al. [20,23], de Chiusole et al. [21,24] and Robusto et al. [22]. Within CbKST, Competence-based Test Development (CbTD) [26,27] is a novel method that can be used to construct a test from scratch, as well as to improve or shorten an existing test. CbTD allows obtaining assessment tools that are as informative as possible about the competence state underlying the item responses. If desired, the test can be minimal, meaning that all the items contribute to test informativeness. In CbKST, the test can be represented by a *competence model* [66] that associates, to each item, all the skills assumed to be relevant to solving it. CbKST is not the only framework for assessing the skills an individual masters from the observed item responses. For instance, Cognitive Diagnostic Models [67,68] represent a valuable alternative. It is worth mentioning that there is a close correspondence between the latter framework and CbKST, which ranges from very basic concepts to formal models [69].

In the framework of CbKST, the CoTeSt consists of a total of 20 items, where the collection of skills needed to solve them consists of the skills identified in section Assessing

Collaborative and Teamwork Skills in Teachers. The association between items and skill is depicted in Fig 1.

As can be seen, each item is associated with the skill(s) that an individual has to master in order to solve it, according to different solution strategies. For instance, Item 1 requires only skill 'Identifying and planning action strategies' to be solved. Some items are repeated in two different rows and are associated with two different skills. This means that there are two strategies for solving these items, each involving one of the two skills. This is the case of Items 4, 6, 11, 13, 14, 16, 19. For instance, Item 4 needs either skill 'Monitoring performance and

| ITEMS | SKILLS | | | | | |
|---|---|---|---|---|---|---|
| | Using proactive communication strategies | Using active listening techniques | Recognizing type and relevance of conflict situations | Identifying conflict resolution strategies | Identifying and planning action strategies | Monitoring performance and providing feedback |
| 1 | | | | | X | |
| 2 | | | X | | | |
| 3 | | | | | X | |
| 4 | | | | | | X |
| 4 | | | | | X | |
| 5 | | | X | X | | |
| 6 | | | | | | X |
| 6 | | | | | X | |
| 7 | X | | | | | |
| 8 | | | X | X | | |
| 9 | | X | | | | |
| 10 | | | | | | X |
| 11 | X | | | | | |
| 11 | | X | | | | |
| 12 | X | X | | | | |
| 13 | | | | X | | |
| 13 | | | X | | | |
| 14 | | | X | | | |
| 14 | | | | X | | |
| 15 | | | | | | X |
| 16 | | | | | X | |
| 16 | | | | | | X |
| 17 | | | | | X | X |
| 18 | | | | X | | |
| 19 | | | | | X | |
| 19 | | | | | | X |
| 20 | | | | | | X |

**Fig 1. Association between items and skills.** Each item is linked with the skill(s) relevant to solve it. For a fixed item, the two skills on the same row are required to solve the item, whereas each of the two skills on different rows is sufficient to solve it.

providing feedback' or skill 'Identifying and planning action strategies' to be solved. Other items are linked with two skills in the same row, both of which are required to solve the item; thus, there is only one solution strategy for these items, involving the mastering of both skills. This is the case of Items 5, 8, 12, 17. For instance, Item 5 requires both skills 'Recognizing type and relevance of conflict situation' and 'Identifying conflict resolution strategies' to be solved. These associations between items and skills, as defined in the competence model, represent a theoretical model that needs to be validated on empirical data. A probabilistic model that can be used to this end is the Competence-Based Local Independence Model (CBLIM) [69, 70].The absolute goodness-of-fit of the model to the data can be assessed using Pearson's Chi-square statistic. Goodness-of-fit is satisfactory if the *p*-value of the Pearson's Chi Square statistic is >.1. The CBLIM takes into account two probabilities estimated for each item, defined as careless error and lucky guess. The former is the probability that an individual fails the item despite mastering all the skills needed to solve it, whereas the latter is the probability that an individual solves the item despite lacking some or all the required skills. Possible reasons for these outcomes might be related to the properties of the items. For instance, the wording of the item could confuse the respondents, leading them to provide an incorrect response. In the case of multiple-choice items, respondents could guess the correct answer. Low values (<.5) for the estimated careless error and lucky guess probabilities of an item indicate that there is agreement between the responses an individual is expected to give, based on the skills they master, and the responses that the individual actually gives.

Once the CBLIM has been validated on empirical data, it can be used to uncover the set of skills an individual masters (called the *competence state*) from the observed item responses. Among all possible competence states that could characterize the population of individuals, the one with the largest likelihood is chosen to represent the uncovered competence state of the individual. The higher the likelihood value of this competence state, the greater the confidence that the skills the individual masters have been correctly identified [71]. The empirical validation of the CoTeSt is described in section Quantitative Analysis.

## Results

### Test design

The CoTeSt was designed to capture and maintain participants' interest by engaging them in typical role-play scenarios. It consists of a total of 20 sequential items, in both multiple-choice and short answer formats. Each item is associated with one or more skills, as depicted in Fig 1, and responses are dichotomously scored as either correct or incorrect. Test validation was carried out using both qualitative and quantitative methods to check whether items assess the hypothesized skills, as detailed in section Test Validation.

In the CoTeSt, teachers are asked to virtually interact, make decisions both individually and collectively, and participate in various activities with colleagues, represented by conversational agents named: Lia, Mirta, and Armando. These names were chosen because they are uncommon in Italy, minimizing potential biases that could arise if participants associated the names with personal experiences.

**Assessment scenario.** Items were designed with an overview of the dimensions where collaborative and teamwork skills can be applied in teachers' daily professional practice. Indeed, the considered skills do not emerge spontaneously but require a structured scenario. For instance, Dillenbourg [72] states that collaboration requires a symmetrical relationship among participants in terms of status, knowledge and goals. However, task types within the team could be very different, influencing perceived status. Moreover, debate within a group can be created by providing members with incomplete information (*jigsaw task*) to achieve a

common goal or by assigning individual goals to each participant [73]. In the framework of the assessment, it is impossible to test all the possible combinations of the factors that may affect the mastery of these skills in a specific context. Therefore, Table 2 presents a schematic overview of the assessment scenario, detailing the situational context in which the assessed skills are applied.

As shown in the table, the assessment scenario consists of five core dimensions: environment, group composition, role of members, medium format, and task content. The scenario is set in a school environment, reflecting the context of formal teaching. Teachers are required to demonstrate their skills in various interactional settings, as indicated by group composition: individually, in pairs, in groups of four, and in a classroom setting. Within these interactions, the roles of members are designed to be both symmetrical (i.e., interacting with colleagues) and asymmetrical (i.e., hierarchical relationships with headteachers or students). The medium format integrates technological tools, including chat-based, video-based, and email-based interactions, to emulate authentic professional communication. The task content outlines a wide range of teaching practices (selected based on TALIS findings [8]) that require teachers to apply their skills in realistic professional contexts.

An item of the CoTeSt requires the teachers belonging to the 'continuity' group work to choose the didactic experience that different classes have to develop in order to participate in the national call. To achieve this, the teachers are immersed in an interpersonal context, in which the team is composed of four members (Lia, Mirta, Armando, and the respondent), in a symmetrical status due to the fact that they are colleagues. Finally, in order to reach a consensus, the teacher in front of the screen interacts with the conversational agents through a simulated chat.

**Software & tools.** The test was developed using Typeform [74], chosen for its flexibility of layout and ease of data collection, analytics, and reporting. In fact, it is a web-based platform that can be used to create both surveys and apps, to collect data, feedback and more, without the need for programming skills. The test was built from scratch, without using the templates offered by the tool. The starting point was represented by the creation of the 20 items, which involved selecting the multiple-choice item types (e.g., picture choice) and the short answer formats (e.g., short text, statement). By using the Typeform feature to structure logic paths,

**Table 2. Dimensions of the assessment scenario considered in the CoTeSt.**

| Dimensions | States |
|---|---|
| Environment | Formal teaching in school |
| Group composition | Individual |
| | Pairs |
| | Groups of four |
| | Classroom |
| Role of members | Symmetrical (colleagues) |
| | Asymmetrical (headteacher, students) |
| Medium format | Chat-based interactions |
| | Video-based interactions |
| | Email-based interactions |
| Task content | Planning lessons in teams |
| | Engaging in cross-class activities |
| | Providing feedback to both students and colleagues |
| | Dealing with students |
| | Participating in school-wide collegial activities |
| | Sending formal emails to the headteacher |
| | Discussing work with school colleagues |

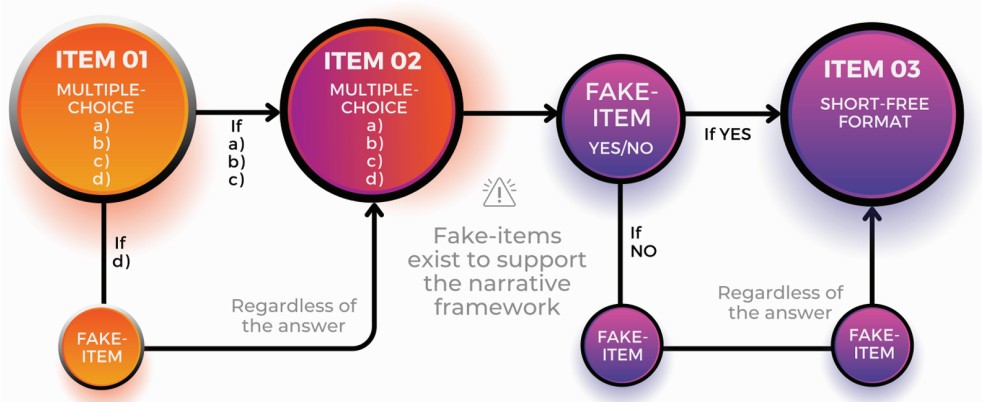

**Fig 2. Control flow design.** Illustrative control flow example for monitoring the order of item presentation. Fake-items are introduced based on responses to ensure a coherent progression of the narration and the same order for assessment items to all users.

the design of the assessment tool follows a specific control flow to manage different answers so that respondents navigate through the test items in the same order. As an example, the control flow applied to three illustrative items is depicted in Fig 2. This helps clarify the functionality and necessity of fake-items. The fake-items, in fact, were introduced to guide participants through the narrative without disrupting the flow of the conversation and to allow them to explore all the alternative paths without influencing their genuine behaviors. In fact, responses to fake items were not analyzed.

As shown in Fig 2, Item 01 is structured as a multiple-choice question with four alternatives. If users select options a), b) or c), they will proceed to Item 02. Otherwise, if they select option d), they will encounter a fake-item. Regardless of the selected option, users will be directed to Item 02. The design of fake-items was essential to maintain coherence in the narrative progression and to ensure a consistent experience for all users. In other words, the test items are presented in a fixed order to all participants. Subsequently, users will be guided through the fake-item that requires a yes/no response and again their selection choice will determine the path to arrive to Item 03. If users select 'yes', they will be directed to Item 03, otherwise they will come into two additional fake-items which are necessary to support the narrative framework according to their previous answer, before being directed to Item 03.

The display was designed to simulate a chat-based message exchange. Therefore, each slide contains an image on the left side, where conversational agents appear to interact with one another. The goal was to make the chat conversations as real as possible. Images were created with Visme [75], a graphic design tool that allows for custom colors, images, and icons to create phone-style screenshots, giving the appearance of regular chat displays. The final result counts 34 slides, not all of which contain assessment items; in fact, some of them are simply narrative elements that advance the story or fake-items. Last but not least, aesthetic care in the display design was an essential part of tool development, as it was necessary to help teachers feel comfortable with the computer-based task. This motivated the choice of chat images, with custom colors in various shades of green, as well as the presence of a welcome and closing slide with a series of soft-coloured icons, very familiar to the school context. Fig 3 provides three illustrative screenshots of the graphical user interface. In the first image, conversational agents appear as colleagues through their accounts on a social platform. They are introduced

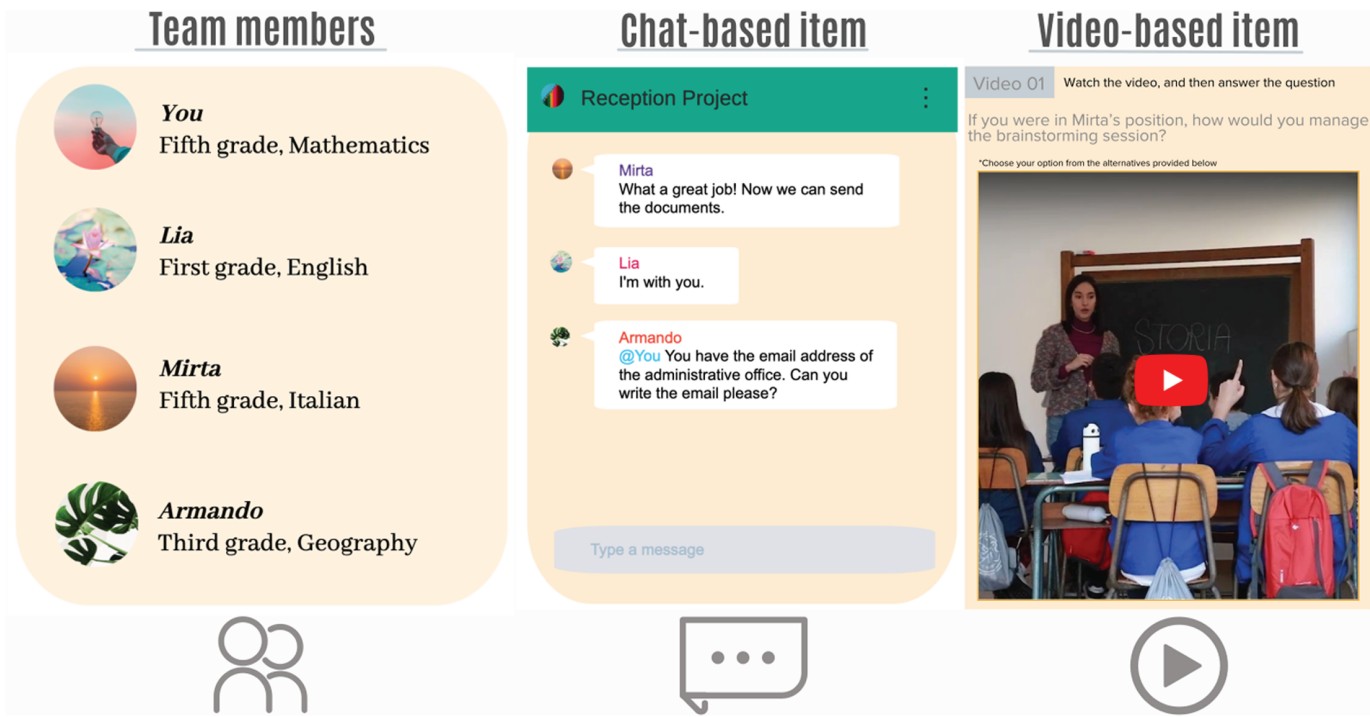

**Fig 3. User interface elements.** Three illustrative examples of the graphical user interface. Conversational agents appear as colleague accounts with unconventional names. Items are generally presented in a chat-based format or as a video to simulate specific educational contexts.

to users by their name, school subjects, and teaching grades. The second screenshot shows an example of a chat-based item, in which conversational agents message each other in a group chat. The user has to answer in the chat by selecting one of four provided options displayed alongside the chat. The third image presents a video-based item, where users have to watch the video and then answer a question by choosing from four alternatives, as indicated in the instruction above the screenshot.

### Experimental campaign

**Procedure.** Data collected from a sample of 139 teachers from preprimary and primary Italian schools were analyzed to validate the CoTeSt within the framework of CbKST (details are provided in section Quantitative Analysis). The first 99 teachers (hereafter referred to as Zoom Group) compiled the online test after the approval from their school headteacher. On a designated day, the teachers joined a Zoom session with the experimenters. The study was presented as a research project aimed at improving the teaching profession. By clicking on the link in the Zoom chat, teachers accessed and completed the test. Afterwards, individual interviews and a group discussion were conducted to explore how teachers perceived the assessment activity. Volunteers were interviewed individually in a private Zoom room immediately after completing the test. Interviews followed a semi-structured format with three questions, covering the following topics: a) experience, b) digital interface and interaction, c) perception. The average interview time was 6 minutes. Teachers' responses were collected, noted and analyzed using Thematic Analysis [76], which enables systematic organization of results, further discussed in section Participants' feedback. As teachers completed the interviews, they returned to the main Zoom room, where they were invited to share their reflections in a group

discussion to identify connections between the activity and their professional practice (details are provided in section Participants' feedback). After gathering sufficient feedback, the test was made available as a link for offline completion on computers and smartphones. This version allowed responses to be collected from an additional 40 teachers (hereafter referred to as FreeLink Group).

The study was approved by the Ethical Committee for Psychological Research of the University of Padua (protocol n. 382-a, approved on January 9, 2024). Data collection took place from February 21 to July 5, 2024. Teachers provided written informed consent to their participation in the study.

**Sample.** The total sample of 139 Italian school teachers includes 129 females (93%) and 8 males (6%), with 2 individuals not reporting their gender. Participants' ages ranged from 25 to 65 years ($M$ = 42.3, $SD$ = 12.9). Of the sample, 122 teachers (87.7%) worked in primary schools, while 17 (12.3%) were employed in preprimary schools. Overall, most teachers were based in Piedmont, followed by Sicily. The sample appears to reflect the demographic composition of the Italian teaching profiles in terms of age and gender. Indeed, according to the Italian Ministry of Education and Merit [77], women make up more than half of the teaching workforce. Specifically, in primary education contexts, 91.3% of teachers are female. This gender imbalance is observed globally, but three countries stand out, namely Latvia, Israel, and Italy [78]. The latest collection of demographic data on the Italian teaching workforce [78] shows a mean age of 49 years ($SD$ = 10.1), which includes teachers from primary to upper secondary levels.

An interesting aspect of the sample is related to the years of service of the teachers (i.e., Seniority). As previously mentioned, the sample consists of two groups: the Zoom Group and the FreeLink Group. Fig 4 *(left)* shows the distribution of participants' years of service, expressed as percentages, across four experience ranges [1–10], [10–20], [20–30], and [30–40] years of service. The data show a clear trend, with most teachers in both groups reporting few years of service. Notably, the 1-10 years range includes 45.5% of participants in the Zoom Group and an even higher 77.5% in the FreeLink Group, indicating a predominance of relatively early-career teachers, especially in the FreeLink Group. In contrast, more experienced teachers (with more than 10 years of service) are less represented in both groups, with the percentage gradually decreasing as years of service increase. A closer look at the distribution within the initial 10-year range is provided by Fig 4 *(right)*, which highlights that the sample is predominantly composed of teachers with 1 to 5 years of service in both groups. Specifically, 29.3% of the Zoom Group falls in the 1–5 year range, compared to the 62.5% in the FreeLink Group, indicating a higher concentration of early-career teachers in the FreeLink Group. Conversely, the 5–10 year range includes a smaller proportion of participants in both groups (16.2% for the Zoom Group and 15% for the FreeLink Group). This detailed examination confirmed the overall trend of relatively limited years of service among participants, particularly within the FreeLink Group.

**Test validation.** The competence model of the CoTeSt was validated using both qualitative and quantitative methods to ensure empirical robustness and conceptual coherence.

**Qualitative analysis.** A qualitative method was used to check whether the items assess the hypothesized skills, as depicted in the item-skill assignment in Fig 1. Three experts, primary school teachers from two different schools in Turin (Italy), were asked to read the test and discuss the clarity and vocabulary of items and answers. Individual sessions were conducted, during which each teacher provided specific feedback to guide further refinements of the test. The final version of the test was the result of this process and was approved by all the teachers involved. The same experts, along with three additional teachers, were provided with the list of skills and were then asked to identify the skill(s) that, according to their expertise,

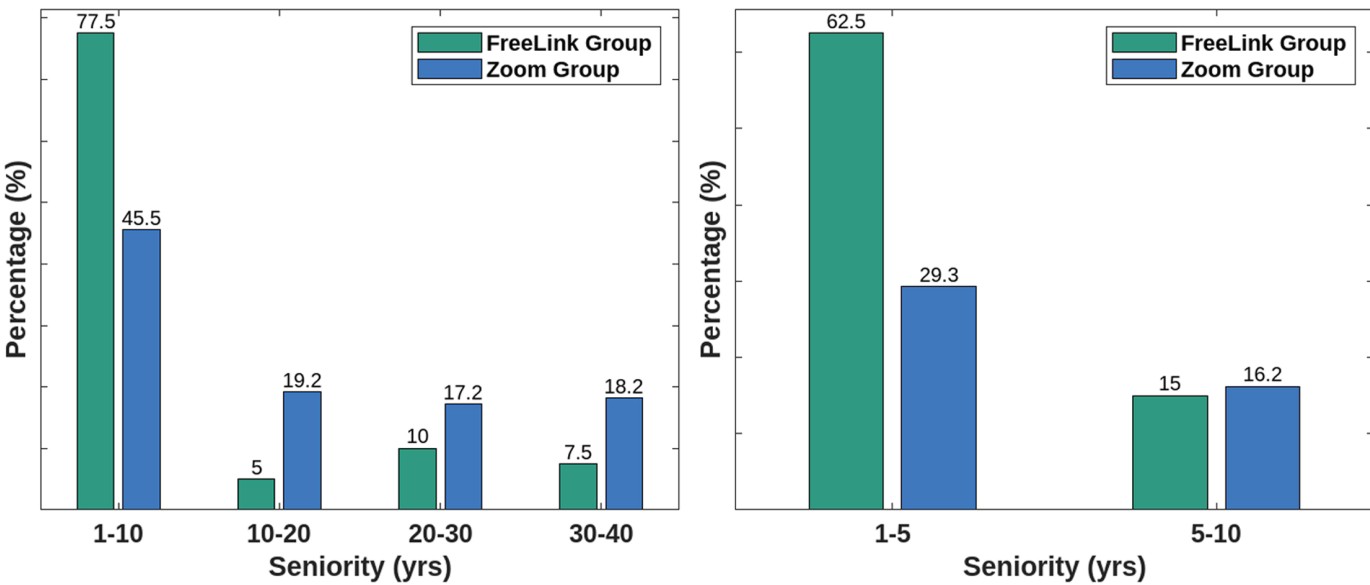

**Fig 4. Distribution by years of service. (left)** Percentage distribution of Zoom Group and FreeLink Group participants across four service ranges: [1–10], [10–20], [20–30], and [30–40] years of service. Note that the sum of percentages exceeds 100 due to rounding of the individual values. **(right)** Detailed percentage distribution of participants with 1–10 years of service in the Zoom Group and FreeLink Group, further divided into two sub-ranges: 1–5 and 5–10 years of service.

were required to correctly answer each item. Their evaluations confirmed the hypothesized item-skill assignment.

**Quantitative analysis.** As outlined in section Competence-based Knowledge Space Theory, the competence model behind the assessment tool needed to be validated using empirical data. For this purpose, the data collected from the sample of 139 teachers were analyzed. Each item was thought to require particular skills to be solved, as depicted in Fig 1. However, this does not ensure that respondents actually mobilize the hypothesized skills to solve the items (for instance, they could employ alternative solution strategies beyond the hypothesized skills). Therefore, administering the test is necessary to determine which skills are effectively used by individuals when responding to the items. The validation of the competence model on empirical data provides evidence that supports the hypothesized association between items and skills. To this end, the absolute goodness-of-fit of the CBLIM was assessed by comparing the observed response patterns of teachers (i.e., the responses provided by the teachers to the items, coded as correct/incorrect) with the expected response patterns derived from the item-skill assignment in Fig 1. Pearson's Chi-square statistic showed that the empirical data did not significantly deviate from the expected ones (*p*-value = .37), indicating that the goodness-of-fit is satisfactory. Moreover, the mean of the careless error probabilities is .21, whereas that of the lucky guess probabilities is .37. These results suggest that the item-skill assignment in Fig 1 appropriately describes the skills required for solving each of the test item [28,79].

The analyzed sample included participants who compiled the test via the Zoom platform and those who accessed it through the free link. We decided to merge the data from the two groups to ensure that the analysis reflected the response behavior of the participants, regardless of the platform used. This decision is further supported by the analysis of a common variable, such as completion times, which could provide insights into teachers' attentiveness and care while completing the test. Fig 5 shows the comparison of completion times between the

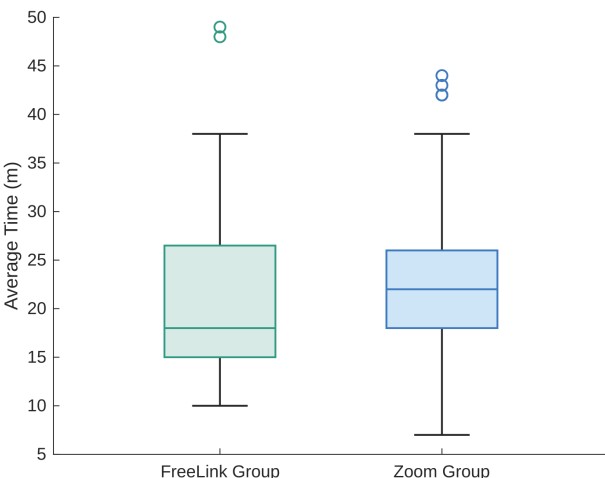

**Fig 5. Time comparison.** Boxplots of completion time (in minutes) for the Zoom Group and the FreeLink Group.

Zoom Group and the FreeLink Group. Specifically, the Zoom Group spent an average of 23 minutes completing the test ($SD$ = 7; $Median$ = 22), whereas the FreeLink Group spent an average of 22 minutes ($SD$ = 10; $Median$ = 18). Outliers were not considered problematic, as some assessment steps (e.g., reading a supplementary optional document attached to the display) required additional time. The results show no statistically significant difference (with Type I error probability $\alpha$ = .01) in the time spent on the test, regardless of the mode of compilation ($t$ = .58; $df$ = 137; $p$-value = .56), suggesting comparable levels of engagement and attention between the groups. Nonetheless, the comparable completion times observed in both groups could indicate a consistent level of engagement and attention, thus justifying the aggregation of data into a single dataset.

**Participants' feedback.** The findings from the interviews are presented in the following sections, organized according to the three main topics (experience, digital interface and interaction, perception), followed by the outcomes that emerged from the group discussions. Key points from the teachers' interviews are shown in '*italics*' throughout the text.

**Experience.** Feedback from the interviews highlighted that the progression of the narrative was highly effective, allowing teachers to become fully immersed in the story, due to their active participation in responding to the items. They acknowledged the activity as a '*formative experience*' for teachers' professionalism, with the possibility to focus attention on behaviors that typically occur within the school context. The narrative framework fostered an environment where teachers did not perceive themselves as being evaluated at any point, nor did they consider the activity an assessment. Overall, teachers described the experience as '*unexpectedly enjoyable, engaging and motivating*'. This is supported by the spent and perceived time: participants spent an average of 22 minutes completing the test but reported that they perceived it to take only about 10 minutes.

**Digital interface and interaction.** Teachers described the display interface as '*user-friendly and intuitive*', and stated that it guided the development of the narrative in a clear way. The presence of conversational agents was perceived as natural and posed no issue. Indeed participants referred to them as '*colleagues*', without questioning their artificial nature or pre-programmed responses. Their intervention in the story was described as '*challenging*', because everyone had their own concepts and beliefs, making it like '*dealing with real-life colleagues*'.

However, unlike familiar colleagues, teachers did not know them well, making it difficult to anticipate their responses or plan the best way to engage with them. Thus, teachers described their responses as '*spontaneous*'. Additionally, teachers focused their attention on the didactic contents of the activity, providing researchers with reflections and insights about the main topic (specifically, the participation in the national call for proposals), the contents of the items from a pedagogical perspective, and the behaviors of students in the videos. These outcomes suggest the effectiveness of the realistic setting of the assessment tool and the ease of interface usability.

**Perception.** Teachers' responses focused on the pedagogical approaches employed by the conversational agents or on the educational implications of specific scenarios presented in the videos. Teachers' reflection confirmed that they were completely immersed in the narrative framework, perceiving the activity as a professional development opportunity rather than merely as an evaluation. Furthermore, after a few minutes of discussion, teachers identified the key value of the experience as lying in '*the interpersonal dynamics among colleagues and their impact on educational practice*'. In this regard, all teachers mentioned 'collaboration' as one of the most important aspects of professional growth, emphasizing how the lack of time often prevents them from focusing on it during their work routines. Their findings suggest that the assessment experience was perceived as an opportunity to engage in typical social interactions, helping them become aware of the significance of developing appropriate social skills and attitudes to improve their professionalism and, consequently, student outcomes.

**Group discussion.** During the group discussions, teachers emphasized the realistic environment in which they were immersed, perceiving no differences between the outlined scenarios and their usual professional experiences. By interacting with others, teachers identified potential response errors due to the presence of the most socially desirable answers among the alternatives. They justified their own responses by affirming that they were spontaneous, acknowledging that a potential second test administration might have captured revised responses. As the discussions progressed, each group reached a consensus that the main topic of the activity was collaboration in the school context, encompassing both symmetrical and asymmetrical relationships. Some participants also recognized the two main domains assessed (*relationship* and *work management*), describing these as '*something that has to do with our social interaction and something that has to do with our practice*'. In general, group discussions underscored the formative value of the assessment tool, fostering teachers' awareness of their collaborative and teamwork skills and inspiring a desire to further develop these skills for the benefit of their profession and the results of their students.

## Discussion and final remarks

The paper presented the CoTeSt (Collaborative and Teamwork Skill test), which is a test to assess the collaborative and teamwork skills of preprimary and primary school teachers. CoTeSt denotes both the items the test consists of and the real-life scenario around which it was designed. Furthermore, the paper introduced an innovative approach to the formative assessment of the considered skills. By addressing key challenges in designing such an assessment tool, we developed a test from scratch with specific features that accurately assess teachers' soft skills. The collaborative and teamwork skills selected for the CoTeSt are among the fundamental ones for fostering a culture of collaboration within schools, as well as for influencing classroom management and student engagement. The scenario-based assessment was designed to replicate real-life situations that teachers frequently encounter, such as mediating conflicts among colleagues, coordinating group activities, and selecting specific pedagogical

strategies in team settings. The concrete application of these skills in teachers' professional practice ensures an authentic assessment experience.

The findings reveal several promising aspects of the online test, particularly its ability to engage teachers in a non-evaluative environment that encourages authentic responses. This approach allows for a more precise description of their teamwork and collaborative skills. Furthermore, the formative value of the test was confirmed during the group discussion phase, where teachers leveraged the platform to promote a focus group about the current state of professional development and the pivotal role of collaboration in schools. These insights open new avenues for further refinements of the assessment tool, enhancing its applicability and flexibility across educational settings. For instance, the CoTeSt was implemented with scenario-like visuals, chat, video, and email elements. Undoubtedly, a possible next step could be to develop a customized computer-based application that simulates real-time chat interactions among agents. This would create a fully immersive, interactive and practical assessment experience, enhancing the tool's ability to assess collaboration and teamwork skills more authentically.

The CoTeSt holds significant potential for implementation across all educational levels, from preprimary to higher education, and in diverse school contexts, reflecting a variety of social, economic, and cultural backgrounds. Additionally, its adaptability could be enhanced through translation, enabling cross-country applications and fostering comparative analyses of soft skills development over time, across regions, and within the framework of specific educational policies that regulate teacher autonomy and professional training. By fostering authentic self-reflection and professional dialogue, the test has the potential to influence professional development practices across educational contexts, promoting a culture of continuous growth in soft skills. As social skills become increasingly critical in today's education and modern workforce environments [80], this tool represents a meaningful step toward empowering educators and enhancing their collective impact.

Last but not least, the sample reveals that most participants in our study were teachers with only a few years of service. This may indicate that teachers in the early stages of their careers—probably influenced by recent advances in higher education and continuous training policies—are more inclined to engage with innovative educational practices, such as those illustrated in this study. This trend could reflect the enthusiasm of early-career teachers and/or the residual computer illiteracy of more experienced teachers. If the latter is the case, there is a need to implement targeted re-skilling initiatives for more experienced teachers. Equipping all educators with the relevant skills to navigate evolving educational paradigms and social changes is essential for fostering a cohesive, adaptive, and future-ready teaching workforce.

## Acknowledgments

The authors would like to express their gratitude to the teachers who participated in the present study. The first author would like to extend her thanks to the FISPPA Department of the University of Padua and the ISTC of the National Research Council for their support, which significantly contributed to the successful completion of this research.

## Author contributions

**Conceptualization:** Federica Morleo, Pasquale Anselmi, Alessandra Vitanza.

**Data curation:** Federica Morleo, Alessandra Vitanza.

**Formal analysis:** Federica Morleo, Pasquale Anselmi, Alessandra Vitanza.

**Investigation:** Federica Morleo.

**Methodology:** Federica Morleo, Alessandra Vitanza.

**Project administration:** Pasquale Anselmi, Alessandra Vitanza.

**Resources:** Federica Morleo.

**Validation:** Pasquale Anselmi.

**Visualization:** Alessandra Vitanza.

**Writing – original draft:** Federica Morleo, Alessandra Vitanza.

**Writing – review & editing:** Pasquale Anselmi, Alessandra Vitanza.

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
