## [Decision Letter · Decision Letter 0]

1 Feb 2025

PONE-D-24-53150Interactive, Computer-based, and Situated Design for Innovative Formative Assessment ApproachesPLOS ONE

Dear Dr. Morleo,

Thank you for submitting your manuscript to PLOS ONE. After careful consideration, we feel that it has merit but does not fully meet PLOS ONE’s publication criteria as it currently stands. Therefore, we invite you to submit a revised version of the manuscript that addresses the points raised during the review process.

We look forward to receiving your revised manuscript.

Kind regards,

Michael Flor

Academic Editor

PLOS ONE

Journal Requirements:

4. We are unable to open your Supporting Information file Fig1.eps, Fig2.eps, Fig3.eps, Fig4.eps, and Fig5.eps. Please kindly revise as necessary and re-upload.

5. We note that Figure 3 includes an image of a [patient / participant / in the study]. 

7. Please remove all personal information, ensure that the data shared are in accordance with participant consent, and re-upload a fully anonymized data set. 

Additional Editor Comments :

Please read and carefully consider the reviewer comments.

In addition, please consider the following comments from the editor.

[1]

The introduction section mentions (line 21) “teacher collaboration and teamwork”. With whom are teachers collaborating? With other teachers? Make it explicit in the introduction.

[2]

Line 26 “these skills are especially relevant”. That paragraph provides no description of what skills are considered. It is very non-specific. Maybe provide some specific examples? Especially for primary school environments. There are some much later, in Table 1, but even those are very abstract.

[3]

Section “Situated Action”, lines 218-248.

This section essentially describes a test in a simulated environment, which is also a scenario-based assessment. Consider using those terms (“simulated environment” and “scenario-based assessment”), as some readers might be more familiar with such terms.

[4]

Lines 262-266

“The conversational agents respond in a pre-defined way, independent of the teacher’s

responses. This is supported by OECD [36] findings, which highlight that there are no

significant differences in how individuals perform in a computer-based task when

interacting with computer agents compared to human interactions, especially when

agent communication occurs through on screen messages.”

This is very strange. On a naive interpretation, it means there is absolutely no relation between what the agent says and what the teacher says/does. For example, the agent always gives a canned message, like “you won”. That does not make sense to interlocutors. That would not be a conversation.

Can you provide a better description in what sense the agent/examinee ‘responses’ are ‘independent’ ?

Also, the OECD report meant that examinee overall performance (as indicated by scores) is similar in the different settings. It does not mean that the agent and an examinee can “talk past each other”, or does it?

[5]

Line 306, “short-free formats”. This is not a common term in English. The common terms are “free response”, “constructed response” and “short answer”.

[6]

A major problem with current manuscript is as follows.

The manuscript describes a test, but the test is situated in a simulated environment, and that simulated environment is barely described. The situations and the tasks are not described. Are those class-room tasks with (simulated) children or more administrative tasks, like “participation in a national call for proposal”. Those are very different tasks, different skills, and of different importance to educators. So the contents of the simulated scenario are crucial for understanding the novel test.

But very little is provided. Line 304 mentions “engaging them in typical role-play scenarios.”

Where is a detailed description of such scenarios? In what sense are they ‘typical’?

There re some very brief glimpses, like on lines 309-314, and the rest of the section, up to line 387, and Fig.3.

[7]

Another aspect that remains confusing is this: is the test administered as part of a full scenario-based simulation, or is it more a test-session that is ‘lightly decorated’ with scenario-like visuals and chat pieces? Seems more like the latter (according to lines 391-409).

Overall, the presentation of the test situation and contents is not clear enough.

[8]

Line 459

“The absolute goodness-of-fit”

Please describe in detail what measured variable was fitted to what independent variable. It may seem obvious to authors, but not to readers.

[9]

concerning “test validation” (section “Results of Quantitative Analysis”)

In is absolutely unclear in hat sense the test was ‘validated’. The skills and their mapping to items were predefined. What is the validity notion in this case? In what sense Chi-square statistic is applicable here. Please describe explicitly.

[10]

Regarding merging of data (results) between the Zoom and Freelink groups, based on times of completions. What does that actually mean? That the results of the chi-square analysis represent the whole group? Would the results be different within each group?

[11]

Regarding ages and experience that were presented earlier. There is no analysis by experience groups.

Maybe collapse them if needed.

[12]

Regarding the fact/finding that most of the interviewed teachers were not aware of what actually was measured (lines 517/530). This suggests that the test is a ‘stealth test’, and that rises some ethical considerations if such a test would be used in operational settings. That should be discussed.

[13]

The discussion section should discuss the abstract skills that are supposedly measured in the test and their applicability to real-life situations, their representativeness. Also the notion of validity is unclear. In plain terms: what exactly do you think you are measuring and why are you sure your measures actually measure it.

[14]

Give the structured flow of the test, what is the test reliability? This is not mentioned.

You don’t need to show results on it (if you don’t have any yet), but do discuss this aspect.

Reviewers' comments:

Reviewer's Responses to Questions

**Comments to the Author**

1. Is the manuscript technically sound, and do the data support the conclusions?

Reviewer #1: Partly

2. Has the statistical analysis been performed appropriately and rigorously? 

Reviewer #1: Yes

3. Have the authors made all data underlying the findings in their manuscript fully available?

Reviewer #1: Yes

4. Is the manuscript presented in an intelligible fashion and written in standard English?

Reviewer #1: Yes

5. Review Comments to the Author

Reviewer #1: The manuscript presents a very interesting computer-based assessment procedure to explore different skills that are identified as essentials for pre-primary and primary teachers. Test validation is performed within the Knowledge Space Theory framework. On the one hand, I've found the paper a very interesting reading and I consider the approach followed very promising and engaging. No doubt worth of consideration for publication. On the other hand, I have found the manuscript both a bit fragmented and sometimes not detailed enough. Please let me stress that I consider the following comments more as subjective opinions rather than statements of objective issues in the work, but I still believe that they deserve some consideration on the Authors side.

As to the fragmentation, I was under the impression that splitting the content of the sections "Challenges"' and "Our proposals" was likely meant to clearly separate the literature review part from the original contribution of the Authors, but ended up splitting the reasoning behind the choices of the Authors from the actual conclusions of their reasoning. I had to jump back and forth several times between the sections because the structure was forcing me as a reader to keep in mind everything I read before concerning different aspects and topics related to the project before actually seeing the natural conclusions of the Authors' reasoning for each and every single aspect or topic. As an example, first a literature review on skills is carried out, then several other topics are discussed, only later in the new section conclusions are briefly drawn about which skills should actually be further considered based on the literature review.

As to the lack of details, I had the impression that the Authors might have discussed more in depths the relation and differences of their work with other existing approaches to engage teachers while assessing their skills. I also had the impression that the work sometimes was a bit lacking in technical aspects as if the Authors were avoiding any technical discussion concerning the type of approach and models chosen. Maybe the Authors made this choice to try to engage the broadest possible audience, however while reading the manuscript I had several questions that I felt were basically unanswered. For instance, why KST was chosen? In KST guessing and slipping can only be performed at the item level, vs. for instance a CDM approach in which guessing and slipping might be performed at the skill level. One might wonder if in the context of evaluating teachers' skill it might be more relevant to know with how much confidence we can say that a teacher does actually possess a certain skill rather than assuming that it is there or not and any random effect falls on the items. Similarly, once the KST approach is justified and chosen it might be interesting to provide further considerations on the type of knowledge structure obtained in the assessment and on the meaning, interpretation, and actual values of guessing and slipping parameters involved in the test under the CBLIM.

A few minor comments follow:

1 - lines 9-10. Could the Authors elaborate a little more on the topic and provide literature or data to back their claim up? Not that I disagree with the statement, but I think that the topic deserves a bit more of explanation and underpinning.

2 - lines 94-104. Given its generality and applicability to the other subsections, shouldn't the discussion on the definition of and the relation between skills and competences come before, maybe at the introduction level?

3 -lines 159-160. I am not sure why the (Cambridge?) Center for Evaluation and Monitoring is backed up by a citation to the Sutton Trust-EEF Teaching and Learning Toolkit.

4 - lines 285-287. This is just a quibble, but I found the wording of this sentence slightly confusing. At first glance I thought the Authors wrote `rows' in place of `columns', then by looking at the Figure I realized that they were not giving a Q-matrix/skill map with conjunctive or disjunctive interpretations, but they were actually repeating items in different rows when different solution strategies were present. Maybe the Authors could emphasize that the items are repeated, and maybe also add in Figure some very light separation lines between the different items so that item repetition becomes immediately visible.

5 - lines 360. Could the Authors expand on the role of fake-items and maybe provide an example on how they are needed for the narrative?

6 - lines 433. As the Authors pointed out that the sample roughly captures the gender imbalance of the Italian educational system, I was wondering if also the prevalence of low seniority teachers in both groups is representative of the Italian system or rather to the difficulty in sampling older teachers (e.g., computer illiteracy)? Edit: I see that the Authors elaborate a bit on the topic on the final discussion but I think it would still be better if they addressed it also previously.

6. PLOS authors have the option to publish the peer review history of their article (what does this mean?). If published, this will include your full peer review and any attached files.

Reviewer #1: No

---

## [Author Response · Author response to Decision Letter 1]

6 Mar 2025

Responses to reviewers are uploaded as a submitted file, as suggested in the instruction provided in the decision letter.

---

## [Decision Letter · Decision Letter 1]

6 Apr 2025

Interactive, Computer-based, and Situated Design for Innovative Formative Assessment Approaches

PONE-D-24-53150R1

Dear Dr. Morleo,

We’re pleased to inform you that your manuscript has been judged scientifically suitable for publication and will be formally accepted for publication once it meets all outstanding technical requirements.

Kind regards,

Michael Flor

Academic Editor

PLOS ONE

Reviewers' comments:

Reviewer's Responses to Questions

**Comments to the Author**

1. If the authors have adequately addressed your comments raised in a previous round of review and you feel that this manuscript is now acceptable for publication, you may indicate that here to bypass the “Comments to the Author” section, enter your conflict of interest statement in the “Confidential to Editor” section, and submit your "Accept" recommendation.

Reviewer #1: All comments have been addressed

2. Is the manuscript technically sound, and do the data support the conclusions?

Reviewer #1: Yes

3. Has the statistical analysis been performed appropriately and rigorously? 

Reviewer #1: Yes

4. Have the authors made all data underlying the findings in their manuscript fully available?

Reviewer #1: Yes

5. Is the manuscript presented in an intelligible fashion and written in standard English?

Reviewer #1: Yes

6. Review Comments to the Author

Reviewer #1: I thank the Authors for taking my suggestions into consideration in the new version of the manuscript. I believe that the Authors have addressed most of my concerns in a satisfactory way. Due to my personal interests, I must admit that I would have appreciated some more technical details, but I agree that if the intention is to reach a broader audience these details are not needed. In general, I have found this new version of the manuscript better structured and organized, the flow was more streamlined and topics were properly adressed. Additional details and considerations provided more context and insight. I honestly have no further issues to raise or comments to add so I wish the Authors the best for their publicaton.

7. PLOS authors have the option to publish the peer review history of their article (what does this mean?). If published, this will include your full peer review and any attached files.

Reviewer #1: No

---

## [Editor Report · Acceptance letter]

PONE-D-24-53150R1

PLOS ONE

Dear Dr. Morleo,

I'm pleased to inform you that your manuscript has been deemed suitable for publication in PLOS ONE. Congratulations! Your manuscript is now being handed over to our production team.

Kind regards,

on behalf of

Dr. Michael Flor

Academic Editor

PLOS ONE